# A Flower-like In_2_O_3_ Catalyst Derived via Metal–Organic Frameworks for Photocatalytic Applications

**DOI:** 10.3390/ijms23084398

**Published:** 2022-04-15

**Authors:** Maniyazagan Munisamy, Hyeon-Woo Yang, Naveenkumar Perumal, Nayoung Kang, Woo Seung Kang, Sun-Jae Kim

**Affiliations:** 1Department of Nanotechnology and Advanced Materials Engineering, Sejong University, Seoul 05006, Korea; manichemist@gmail.com (M.M.); hyunket@naver.com (H.-W.Y.); pcnaveenperumal@gmail.com (N.P.); nykang0615@naver.com (N.K.); 2Department of Metallurgical and Materials Engineering, Inha Technical College, Incheon 22212, Korea; wkang651@inhatc.ac.kr

**Keywords:** flower-like In_2_O_3_, metal–organic frameworks, photocatalyst, 4-nitrophenol, methylene blue

## Abstract

The most pressing concerns in environmental remediation are the design and development of catalysts with benign, low-cost, and efficient photocatalytic activity. The present study effectively generated a flower-like indium oxide (In_2_O_3_-MF) catalyst employing a convenient MOF-based solvothermal self-assembly technique. The In_2_O_3_-MF photocatalyst exhibits a flower-like structure, according to morphology and structural analysis. The enhanced photocatalytic activity of the In_2_O_3_-MF catalyst for 4-nitrophenol (4-NP) and methylene blue (MB) is likely due to its unique 3D structure, which includes a large surface area (486.95 m^2^ g^−1^), a wide spectrum response, and the prevention of electron–hole recombination compared to In_2_O_3_-MR (indium oxide-micro rod) and In_2_O_3_-MD (indium oxide-micro disc). In the presence of NaBH_4_ and visible light, the catalytic performances of the In_2_O_3_-MF, In_2_O_3_-MR, and In_2_O_3_-MD catalysts for the reduction of 4-NP and MB degradation were investigated. Using In_2_O_3_-MF as a catalyst, we were able to achieve a 99.32 percent reduction of 4-NP in 20 min and 99.2 percent degradation of MB in 3 min. Interestingly, the conversion rates of catalytic 4-NP and MB were still larger than 95 and 96 percent after five consecutive cycles of catalytic tests, suggesting that the In_2_O_3_-MF catalyst has outstanding catalytic performance and a high reutilization rate.

## 1. Introduction

While contemporary society has progressed at a breakneck pace in recent decades, environmental pollution, particularly wastewater, has emerged as a universal and main threat to human health [1,2]. Organic dyes and nitro compounds, which are discharged into the environment by many industries such as plastic, printing, cosmetics, leather, food, pharmaceuticals, and textiles have received a lot of attention because of their potential carcinogenic and toxicity properties, which have raised environmental concerns [3,4]. As a result, several treatments, such as adsorption, catalytic reduction, oxidation, and degradation have been explored and developed to remove organic pollutants such as 4-nitrophenol (4-NP) and methylene blue (MB) [5,6,7,8,9,10].

The compound 4-nitrophenol (4-NP) is a very poisonous phenolic chemical found in the environment that can harm the central nervous system, kidneys, and blood [11]. Disposal of organic contaminants into potable water bodies diminish the amount of sunlight that reaches the water and thus affects photosynthetic activity [12]. Furthermore, organic dyes are poisonous and carcinogenic, and their treatment methods cannot only rely on biodegradation [13]. Methylene blue (MB), an organic dye used in a variety of sectors including cosmetics, textiles, and leather pollutes the environment [14,15,16]. One of the most successful strategies for eliminating these contaminants from water is photocatalytic dye degradation [17]. Additionally, heterogeneous degradation is an attractive approach for degrading organic pigments due to its low cost and high efficacy. [18].

Due to its superior optical and electrical properties, its nontoxicity, and its great stability, In_2_O_3_ is widely used in a various industries [19,20,21]. The status of In_2_O_3_ properties typically determines the variety in its attributes (crystalline or amorphous). Several methods for obtaining In_2_O_3_ have been documented, including hydrothermal, wet chemical, sol–gel, evaporation, and gas phase synthesis [22,23,24,25,26]. The preparation of In_2_O_3_ with the necessary structure and morphology requires a great deal of technological and scientific effort. However, as nanorods, nanosheets, and nanoflowers of In_2_O_3_ have been described previously, the morphology is responsible for the high catalytic efficiency [27,28,29]. There is a need for easy, efficient, and cost-effective ways to make In_2_O_3_ with the advanced properties that are required. Indium oxide has been widely used in Li-ion batteries, electrochemicals, dye-sensitized solar cells, photocatalysis, and gas sensors [30,31,32,33,34]. Meanwhile, In_2_O_3_ has proven to be an effective sensitizer for extending the absorption spectra of photocatalysts from the UV to the visible range [35]. As a result, research into the photocatalytic properties of In_2_O_3_ has a promising future for the degradation of organic toxins in wastewater and other pollutants in the environment [36,37].

Metal–organic frameworks (MOFs) are utilized as a template precursor to synthesize diverse functional materials utilizing various treatment techniques that result in ordered structures with a wide surface area, regulated pore texture, and high carbon content [38,39]. This method allows for effective control of the catalyst shape and microstructure, which is advantageous for exposing active sites with high surface areas and increasing electron transport in porous structures [40]. Here, for the first time, we describe a simple synthesis approach to prepare flower-like indium oxide (In_2_O_3_-MF), indium oxide microdisc (In_2_O_3_-MD), and indium oxide microrod (In_2_O_3_-MR) catalysts using a MOF based one-step hydrothermal process. The unique structure of In_2_O_3_-MF comprises many voids and more active sites for the photocatalytic process compared to In_2_O_3_-MD and In_2_O_3_-MR. The degradation of MB and reduction of 4-NP in the presence of visible light and NaBH_4_ were used to investigate the photocatalytic activity of In_2_O_3_-MF.

## 2. Results and Discussion

### 2.1. Catalyst Characterization

The synthesis process of the In_2_O_3_-MF, In_2_O_3_-MR, and In_2_O_3_-MD catalysts is presented in Figure 1. Hard Lewis acids and bases, as well as mild Lewis acids and bases, are predicted to form strong coordination bonds [41]. Hard bases, such as carboxylate-based ligands, can create stable MOFs with indium metal cations. Moreover, different acids have distinct effects on the development of different MOF morphologies. Indium nitrate, terephthalic acid, and trimesic acid were utilized as precursors for the synthesis of various In_2_O_3_ MOFs. In_2_O_3_ MOFs particle formation was mediated by two distinct mechanisms: the formation of truncated octahedra or rods and the growth of homogenous pods. In_2_O_3_-MF, a flower-like MOF-based catalyst, is described by a three-dimensional network, large surface area, and it is synthesized via a simple and low-cost hydrothermal reaction using trimesic and terephthalic acids. The FTIR spectra of In_2_O_3_-MF, In_2_O_3_-MD, and In_2_O_3_-MR samples are shown in Figure 2a, and the enlarged FTIR spectrum in 400–1700 cm^−1^ range is presented in Figure 2b. The FTIR spectrum reveals different bands at 400–1700 cm^−1^ because of the vibration corresponding to the main MOF functional groups. The band at 467 cm^−1^ is attributed to the metal–oxygen bond. In the FTIR spectra of In_2_O_3_-MF, In_2_O_3_-MD, and In_2_O_3_-MR samples, the absorption peaks at 3433 and 1622 cm^−1^ are attributed to O–H stretching and bending vibration of physically adsorbed water molecules present on the sample surfaces. The peaks at 715 and 756 cm^−1^ correspond to indium substitution on the benzene groups. The band at 1110 cm^−1^ is related to C–O–In stretching vibrations of MOF. The bands at 1375 and 1440 cm^−1^ correspond to the vibrations of carboxylate groups in TPA and TMA, which is responsible for the bidentate behavior of the COO moiety. The band at 1624 cm^−1^ corresponds to the H–O–H vibration, which indicates that MOF possesses water crystals. The PXRD pattern confirms their crystallinity, MOF structure, and further supports the successful synthesis of In_2_O_3_-MF, In_2_O_3_-MD, and In_2_O_3_-MR as shown in Figure 3. The Bragg diffraction peaks of all samples were identical. The presence of diffraction peaks at 2θ = 17.6°, 18.7°, and 27.1° validated the effective synthesis of MOF using a one-step hydrothermal technique. The results suggested that In_2_O_3_-MR is structurally identical to the MOF parent framework [42,43]. However, when TPA is used as a ligand, the crystalline peaks of In_2_O_3_-MR become wider and weaker than those of other MOFs, but the general characteristics remain the same. At 11.55°, the strongest peak was detected, indicating a significant degree of crystallinity. Some weak, low-intensity peaks were found between 30° and 45°, indicating the presence of In_2_O_3_ phase on a tiny scale. Strong peaks in the XRD pattern indicate that these In_2_O_3_-MDs have good crystallinity, which is supported by the data. However, in the instance of In_2_O_3_-MF, a mixed phase was observed, consisting of cubic and rhombohedral crystals. Reflections from the cubic and rhombohedral In_2_O_3_ Bragg planes (104), (110), (024), (116), and (214) were recorded at 2θ = 30.0°, 33.3°, 46.1°, 51.6°, and 57.4°, respectively. The diffraction pattern is extremely similar to that of In_2_O_3_ (JCPDS file no. 21-0406). Additionally, the crystallinity of the In_2_O_3_-MF is indicated by the distinct and strong peaks. Filed emission-scanning microscopy was used to examine the morphology of In_2_O_3_-MF, In_2_O_3_-MD, and In_2_O_3_-MR catalysts. In_2_O_3_-MD platelet disc form demonstrated the usual FESEM pictures as shown in in Appendix A. As shown in Appendix A, In_2_O_3_-MR exhibited a rod-like morphology. In Figure 4a,b, the SEM images of In_2_O_3_-MF present a three-dimensional (3D) microflower shape with self-assembly nanosheets. In_2_O_3_-MF is shaped in rankly micropetals with the rougher surface (Figure 4c) indicating that the morphology of the catalysts remained intact even after the annealing process. Although the overall shape of the microflower remained almost intact, the micropetals were transferred to be disordered and porous structures. The 3D microflower provides a broad reaction region for reactants and provides support for the mechanical stability of the catalyst. TEM was used to analyze the precise structure of the In_2_O_3_-MF catalyst. In_2_O_3_-MF catalysts are made up of interconnected micropetals with many mesopores within them, as can be observed (Figure 4d–f). The EDS studies of In_2_O_3_-MF revealed that C, O, and In elements coexist in the sample, as shown in Figure 4g–i. Moreover, the three elements exhibit a homogeneous distribution in In_2_O_3_-MF.

N_2_ adsorption–desorption analysis was used to evaluate the porous structure of In_2_O_3_-MF. They have a mesoporous structure, as shown in Appendix A, and their pore size distribution ranges from nanometer to macroscale. The micropore and mesopore distribution, as measured by the HK and BJH techniques, was used to confirm the pore structure [44]. Appendix A summarizes the BET surface area of In_2_O_3_-MF, In_2_O_3_-MD, and In_2_O_3_-MR. When compared to In_2_O_3_-MD and In_2_O_3_-MR, In_2_O_3_-MF has the greatest BET surface area of 486.95 m^2^ g^−1^. As a result, the In_2_O_3_-MF large surface area will be enriched with active sites for photocatalysis. The optical properties of synthesized catalyst samples were analyzed using UV–DRS. The band gaps in the In_2_O_3_-MF, In_2_O_3_-MD, and In_2_O_3_-MR samples were calculated using the Tauc’s equation [45];
(α*h*ν)^n^ = (*h*ν − E_g_)(1)
where α is the absorption coefficient, հν is the energy of the incident photons, E_g_ is the optical band gap, n = 1 and n = 2 for allowed indirect and direct transitions, respectively. It was expected that the In_2_O_3_ samples would follow the direct allowed transition [46,47]. Therefore, we used n = 2 in the Tauc’s equation for calculating the band gaps in the In_2_O_3_-MF, In_2_O_3_-MD, and In_2_O_3_-MR samples (Appendix A). The band gap values corresponding to the In_2_O_3_-MD, In_2_O_3_-MR, and In_2_O_3_-MF samples were 3.72, 3.32, and 3.00 eV, respectively. The flower-like In_2_O_3_-MF band gap value was narrowed compared to In_2_O_3_-MD and In_2_O_3_-MR, which was probably attributed to the formation of impurity states between In and O in the band gap. The XPS spectra for the synthesized catalyst In_2_O_3_-MF in the In 3d, C 1s, and O 1s areas are shown in Figure 5a. The elements O, C, and In are all present in In_2_O_3_-MF. The deconvoluted spectra of C 1s (Figure 5c) can be separated into multiple peaks. The primary peak at 284.5 eV refers to the sp^2^-hybradized carbon, while three shoulder peaks at 285.1 eV, 286.6 eV, and 288.7 eV relate to alkoxy, carbonyl, and carboxylate functional groups, respectively [48]. The functional groups of H–O–H (532.27 eV), In–O (531.0 eV), and carboxylate groups (529.26 eV) are connected to the deconvoluted XPS spectra for O 1s (Figure 5b). The symmetric peaks at 452.2, 453.2, 445.6, and 444.6 eV were deconvoluted from the In 3d peaks (Figure 5d). Peaks associated to In^3+^ (Indium oxide) were found at 453.2 and 445.6 eV, while those connected to In (metallic indium) were found at 452.2 and 444.6 eV.

### 2.2. Photocatalytic Performances

#### 2.2.1. Catalytic Reduction of 4-Nitrophenol

The UV-visible spectra were measured at 30 s intervals to track the development of the catalytic reduction of 4-NP with NaBH_4_. The 4-NP solution has a prominent absorption peak at 317 nm in aqueous medium. However, when freshly synthesized NaBH_4_ was added, the absorption peak shifted from 317 to 400 nm, and the color changed from light yellow to deep yellow, as seen in Figure 6a. This was owing to the production of intermediate 4-nitrophenolate ions. The absorption peak at 400 nm remained constant in the absence of a catalyst, implying that the reduction of 4-NP does not occur in the presence of solely NaBH_4_ in the solution without catalysts due to repulsion between the negatively charged 4-NP and BH_4_^−^ ions. Metal oxides are well known for effectively catalyzing the reduction of 4-NP by facilitating an electron relay system switching from the donor BH_4_^−^ to the acceptor 4-NP. As a result, after adding the In_2_O_3_-MF catalyst to the NaBH_4_ and 4-NP solution, the reduction reaction begins, and the intensity of the maximum absorption peak at 400 nm rapidly decreases, along with the 4-NP solutions rapidly changing color from deep yellow to colorless due to the transformation of 4-NP to 4-AP, which revealed a new absorption peak at 300 nm, as shown in Figure 6b. After 20 min, no recognizable peak of the nitro molecule could be seen at 400 nm, indicating that the reduction of 4-NP was successful. For the reduction of 4-NP over the In_2_O_3_-MF, In_2_O_3_-MR, and In_2_O_3_-MD catalysts, Figure 6c shows the variation of C_t_/C_0_ against reaction time, where C_t_ and C_0_ are the 4-NP concentrations at times t and 0, respectively. As shown in Figure 6c, the photocatalytic efficiency of In_2_O_3_-MD, In_2_O_3_-MR, and In_2_O_3_-MF with a 0.3 g/L dose using the reduction of 4-NP under UV light irradiation after achieving adsorption–desorption equilibrium was 51.98, 59.33, and 99.32 percent, respectively. The achieved results from the photocatalytic study using 4-NP revealed that In_2_O_3_-MF exhibits greater photocatalytic activity as compared to In_2_O_3_-MR and In_2_O_3_-MD, Figure 6d. The flower-like structure of In_2_O_3_-MF was discovered to contribute to the better adsorption capacity and effective separation of the charge carriers (electron–hole pairs), resulting in increased photocatalytic activity. Thus, the In_2_O_3_-MF catalyst’s superior catalytic activity over In_2_O_3_-MR and In_2_O_3_-MD was most likely owing to its large surface area and well-controlled pore structure. Overall, the In_2_O_3_-MF catalyst modifies the shape and electrical characteristics of the catalyst, resulting in increased activity for the 4-NP reduction process. In Appendix A, the current photocatalytic results are compared to previously published results. Finally, the stability and sustainability of the In_2_O_3_-MF catalyst were assessed by performing five cycles of regeneration and reusability studies to examine the photocatalytic reduction of 4-NP. Up to five cycles, about 95 percent photocatalytic activity of 4-NP were found, demonstrating that the photocatalytic performance of In_2_O_3_-MF did not degrade significantly, as shown in Appendix A.

#### 2.2.2. Photocatalytic Degradation of Methylene Blue

The potential catalytic application for degradation of MB to LMB through In_2_O_3_-MF, In_2_O_3_-MR, and In_2_O_3_-MD catalysts was further explored and presented in Figure 7. The degradation process was observed by UV-vis spectrophotometry and the spectra were collected from 200 to 800 nm range at ambient conditions. Methylene blue gave its characteristic absorption peak at 664 nm (Figure 7a). Photocatalytic degradation of MB is shown in Figure 7c by C_t_/C_0_ plotting against t (time). The C_t_ and C_0_ denotes MB initial and zero stage. In_2_O_3_-MR and In_2_O_3_-MD catalyse the MB conversion to LMB up to 32% and 36% respectively (Figure 7c). When In_2_O_3_-MR and In_2_O_3_-MD catalysts were introduced into the reaction solution, up to a 30% decrease in the MB content was seen within 120 s. Further rising in time up to 180 s, the residual MB content remained constant (Figure 7a). This suggests that In_2_O_3_-MR and In_2_O_3_-MD catalysts cannot catalyze the MB conversion to LMB. As shown in Figure 7b, full degradation of MB was achieved using the In_2_O_3_-MF photocatalyst with optimum doses of 0.2 g/L under UV light irradiation for 3 min. The plot shown in Figure 7c indicates that MB can be 99.2% converted into LMB within 3 min. The absorption spectrum shown in Figure 7b for MB reduction indicates rapid decline in the MB peak intensity at 664 nm with increasing time suggesting a reduction of MB dye to LMB and the blue color solution turning colorless. These results thus clearly show that In_2_O_3_-MF is much more efficient in reduction of MB than the In_2_-O_3_-MR and In_2_O_3_-MD counterparts. The stability and reusability of the synthesized In_2_O_3_-MF catalyst was investigated by performing an MB reduction reaction five times with the same catalyst. Even after five cycles, the catalyst demonstrated higher catalytic activity, with an efficiency of over 96%, as shown in Appendix A. The loss of porosity in the synthesized In_2_O_3_-MF could reflect the fall in efficiency rate. Furthermore, in Appendix A, the performance of the In_2_O_3_-MF catalyst for the reduction of MB to LMB is compared to that of other catalysts previously reported in the literature. Appendix A clearly shows that the In_2_O_3_-MF catalyst performed better. The stability and reusability are two crucial factors for the real-time application of a photocatalyst. Furthermore, the structure of the photocatalyst used for five cycles of catalyst was then characterized using SEM and elemental mapping analysis (Figure 8), which revealed no obvious alteration of the particles and was indicative of the high stability of the photocatalyst. Additionally, these results clearly indicated excellent stability of In_2_O_3_-MF over five consecutive cycles for the degradation of MB.

Isopropanol (IPA), triethanolamine (TEOA), and benzoquinone (BQ) were used in a trapping model to eliminate hydroxyl radicles, holes, and superoxide radicles, respectively. Figure 9 shows how deferent scavengers (IPA, TEOA, and BQ) affect the photocatalytic degradation of MB over In_2_O_3_-MF when exposed to UV-vis light. As a result, trapping the ^•^OH radicles with IPA dramatically lowered the photocatalytic activity of the In_2_O_3_-MF sample, demonstrating that ^•^OH radicles play a significant role in photocatalytic degradation of MB to LMB (Equations (2)–(5)). Basically, the photocatalytic mechanism is stated as follows:In_2_O_3_-MF + *հ*ν → In_2_O_3_-MF (*e*^−^) + In_2_O_3_-MF (*հ*^+^)(2)
In_2_O_3_-MF (*e*^−^) + O_2_ → ^•^O_2_^−^(3)
In_2_O_3_-MF (*հ*^+^) + H_2_O → H^+^ + ^•^OH(4)
^•^O_2_^−^ + ^•^OH + pollutants → CO_2_ + H_2_O + intermediate products(5)

Reactive oxygen species (ROS) such as superoxide radical and hydroxyl ions, which are formed as electron hole pairs, are responsible for photocatalytic destruction of organic contaminants [49,50]. The photocatalytic reduction and degradation reactions of 4-NP and MB associated with the generation of charge carriers via UV-light irradiation of a flower-like In_2_O_3_-MF photocatalyst surface resulted in valence bond stimulation, electron transfer to the conduction band (CB), and the creation of the same amount of hole in the valence band. When these photoexcited charge carriers came into contact with dissolved oxygen (O_2_) and water molecules (H_2_O) in 4-NP and MB, superoxide radicals and hydroxyl ions were produced. The ROS reacted with the surface adsorbed 4-NP and MB, converting them to simpler intermediates such as 4-AP, CO_2_, H_2_O, NH^4+^, and so on [51,52]. Figure 10 depicts a schematic photocatalytic reaction mechanism for reduction of 4-NP and degradation of MB using In_2_O_3_-MF as photocatalysts.

## 3. Materials and Methods

All of the chemical reagents and solvents utilized in this investigation were analytical grade and were used straight from the package. Indium(III) nitrate hydrate (In(NO_3_)_3_ xH_2_O), terephthalic acid (TPA; 98%), trimesic acid (TMA; 95%), and potassium hydroxide (KOH; reagent grade) were purchased from Sigma-Aldrich, Seoul, South Korea.

### 3.1. Synthesis of In_2_O_3_-MF

Trimesic acid (420 mg) and terephthalic acid (250 mg) were first dissolved in 50 mL of N,N-dimethylformamide, and then indium(III) nitrate hydrate (980 mg) was added separately to the aforesaid solution. After stirring 2 h, the reaction mixture was transferred to a Teflon autoclave and incubated in an oven at 120 °C under static conditions for 12 h. After the reaction was completed, the sample was filtered, washed three times with D.I water and ethanol, and air dried at 80 °C for 24 h. To obtain the product In_2_O_3_-MF, the flower-like In_2_O_3_ was annealed in an argon atmosphere set at 300 °C with a ramp of 3 °C min^−1^ for 2 h. Using similar procedures, the corresponding In_2_O_3_-MD and In_2_O_3_-MR catalysts were prepared with trimesic and terephthalic acids only.

### 3.2. Physical Characterization

X-ray photoelectron spectroscopy (XPS; K-alpha, Thermo Scientific Inc., Waltham, MA, USA) was used to evaluate the elemental composition and chemical bonding of In_2_O_3_-MF, In_2_O_3_-MD, and In_2_O_3_-MR, and X-ray powder diffraction (XRD) (XRD PANalytical, Almelo) was utilized to investigate the morphology of the synthesized samples. A Bruker (Vertex 70) spectrometer was used to record the FTIR spectra of In_2_O_3_-MF, In_2_O_3_-MR, and In_2_O_3_-MD samples in the region of 4000–400 cm^−1^. A UV 5000 VARIAN Agilent spectrophotometer was used to record the photocatalytic studies and diffuse reflectance spectra of the materials in the UV-visible range. For reactions that required illumination, a Xe lamp (PLS-SXE300) with a long-pass filter (˃420 nm) was used. The scanning electron microscope (SEM; HITACHI, Tokyo, Japan) was used to study the morphology of the produced samples. JEM-ARM200F was used to perform transmission electron microscopy (TEM) and energy dispersive spectroscopy (EDS) mapping in the STEM mode. The surface area, pore size, and pore volume of the synthesized materials were determined using the Brunauer–Emmett–Teller (BET) and Barrett–Joyner–Halenda (BJH) methods, respectively, utilizing the N_2_ adsorption–desorption isotherm on a nano POROSITY-HQ (Mirae Instruments, Seoul, South Korea) equipment.

## 4. Conclusions

The photocatalysts were evaluating for the first time a flower-like In_2_O_3_-MF synthesized by a simple one-step hydrothermal process. The photocatalysts were evaluating using FTIR, XRD, FE-SEM, EDX, TEM and XPS as they were synthesized. The catalytic activity of freshly generated photocatalyst samples was investigated for the photoreduction and degradation of 4-nitrophenol (4-NP) and methylene blue (MB). Because of its porous flower-like architecture and large surface area, In_2_O_3_-MF displayed higher catalytic activity than In_2_O_3_-MR and In_2_O_3_-MD. The synthesized In_2_O_3_-MF catalyst performed well for 4-nitrophenol reduction, and it also had improved catalytic activity and cycling stability. Methylene blue concentrations as high as 30 mg/L can be completely degraded in 3 min in the presence of 0.2 g of In_2_O_3_-MF photocatalyst. Furthermore, after five consecutive cycles, the catalytic efficiency was greater than 96%, demonstrating that In_2_O_3_-MF is highly recyclable. As a result, the findings of this study can be used to improve photocatalysts for use in environmental remediation.

## Figures and Tables

**Figure 1 ijms-23-04398-f001:**
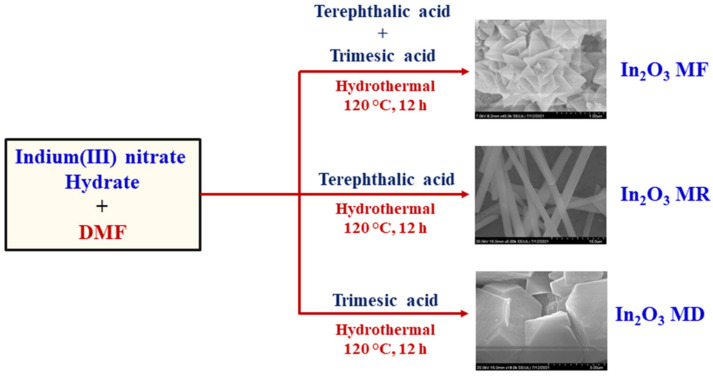
Schematic illustration of the synthesis of the In_2_O_3_-MF, In_2_O_3_-MR, and In_2_O_3_-MD via a hydrothermal process.

**Figure 2 ijms-23-04398-f002:**
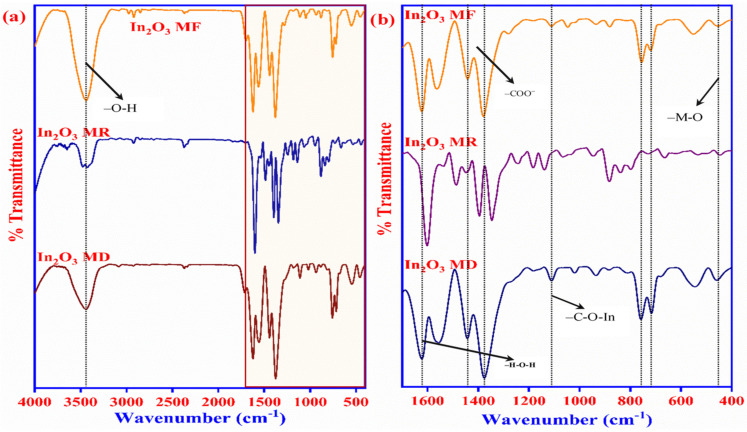
FTIR spectra of the In_2_O_3_-MF, In_2_O_3_-MR, and In_2_O_3_-MD (**a**), Enlarged FTIR spectra of In_2_O_3_-MF, In_2_O_3_-MR, and In_2_O_3_-MD (**b**).

**Figure 3 ijms-23-04398-f003:**
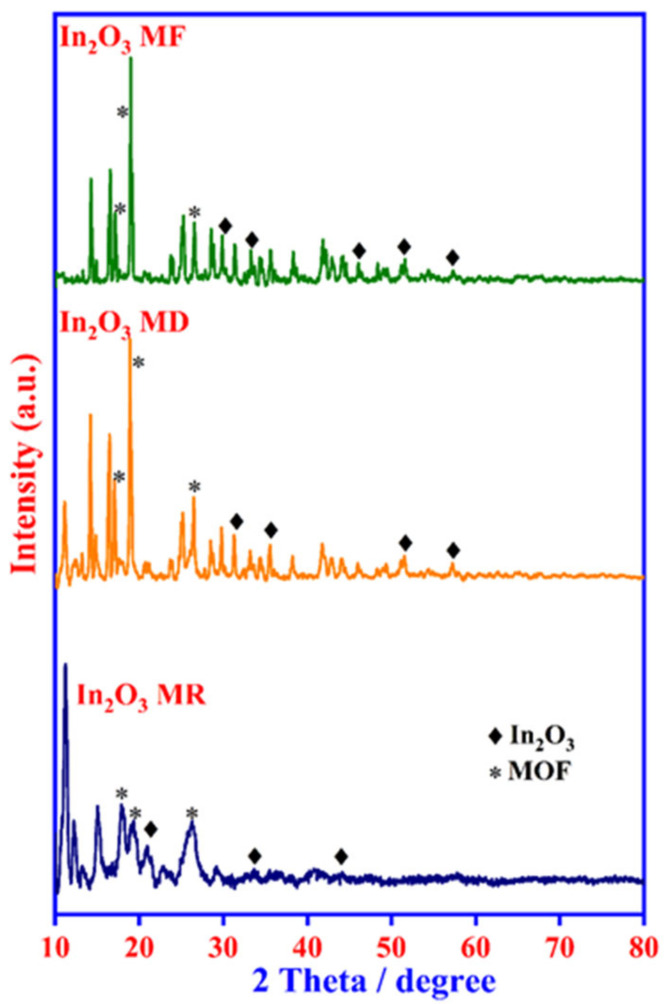
XRD pattern of In_2_O_3_-MR, In_2_O_3_-MD, and In_2_O_3_-MF annealed at 300 °C.

**Figure 4 ijms-23-04398-f004:**
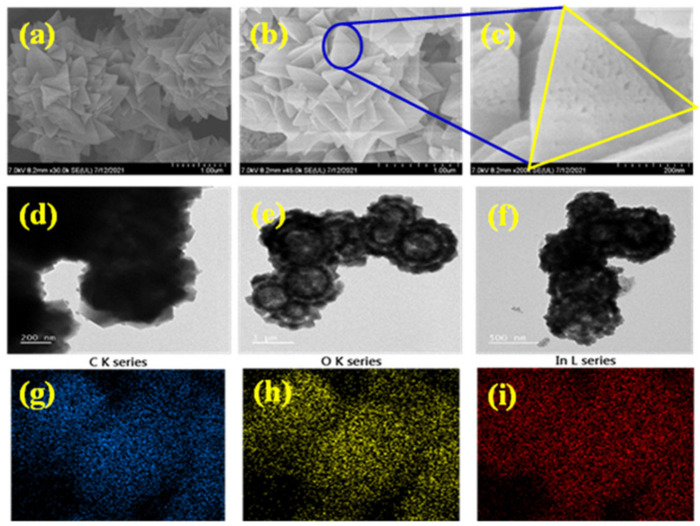
SEM images for In_2_O_3_-MF (**a**–**c**); TEM images for In_2_O_3_-MF (**d**–**f**); elemental mapping images for In_2_O_3_-MF (**g**–**i**).

**Figure 5 ijms-23-04398-f005:**
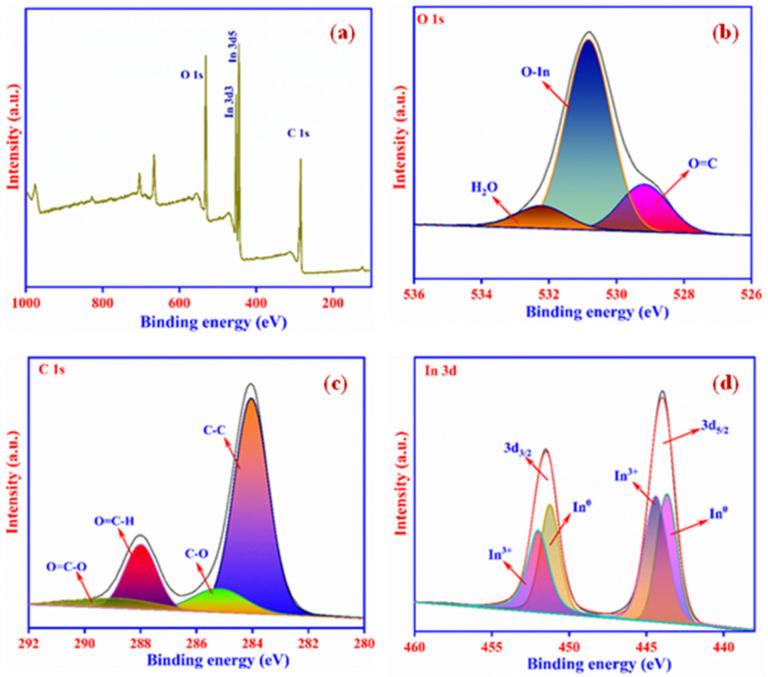
XPS survey spectrum of In_2_O_3_-MF (**a**); XPS spectra and deconvolution of O 1s (**b**); C 1s spectra (**c**); and In 3d spectra (**d**).

**Figure 6 ijms-23-04398-f006:**
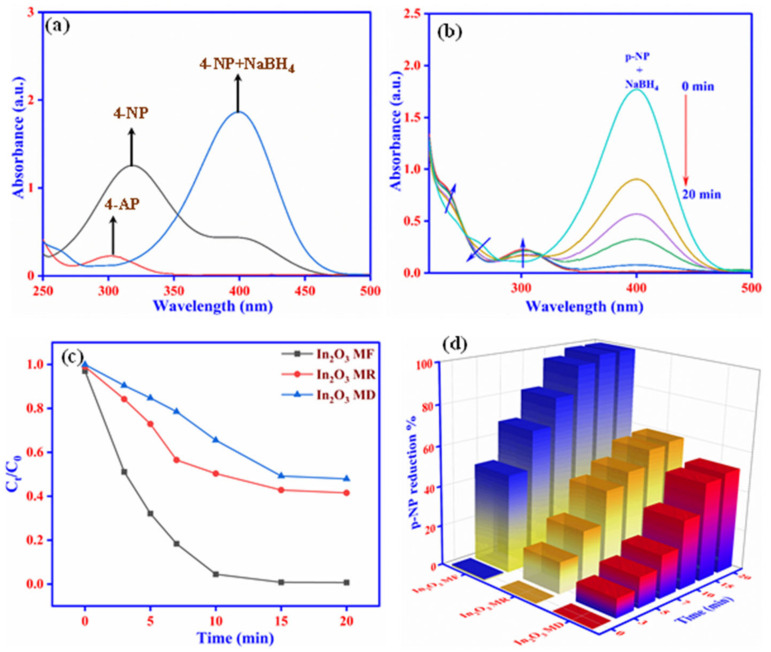
UV-vis spectra of 4-AP, 4-NP, and 4-NP with adding NaBH_4_ (**a**); catalytic reduction of 4-NP using In_2_O_3_-MF as a catalyst (**b**); plots of C_t_/C_0_ vs. contact time using In_2_O_3_-MF, In_2_O_3_-MR, and In_2_O_3_-MD catalysts (**c**), 4-NP reduction % vs. contact time plot (**d**).

**Figure 7 ijms-23-04398-f007:**
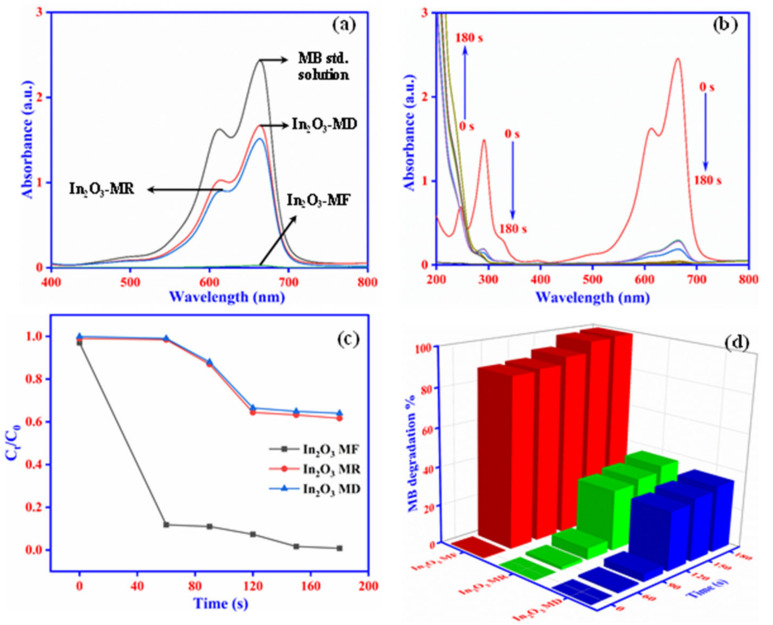
UV-vis spectra of MB degradation with In_2_O_3_-MF, In_2_O_3_-MR, and In_2_O_3_-MD catalysts (**a**); photocatalytic degradation of MB using In_2_O_3_-MF as a catalyst (**b**); plots of C_t_/C_0_ vs. contact time using In_2_O_3_-MF, In_2_O_3_-MR, and In_2_O_3_-MD catalysts (**c**); MB degradation % vs. contact time plot (**d**).

**Figure 8 ijms-23-04398-f008:**
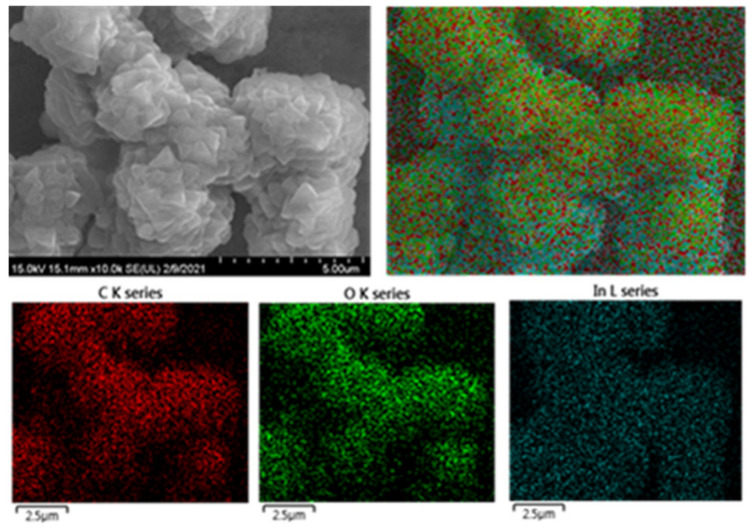
SEM and elemental mapping images of the In_2_O_3_-MF catalyst after five cycles.

**Figure 9 ijms-23-04398-f009:**
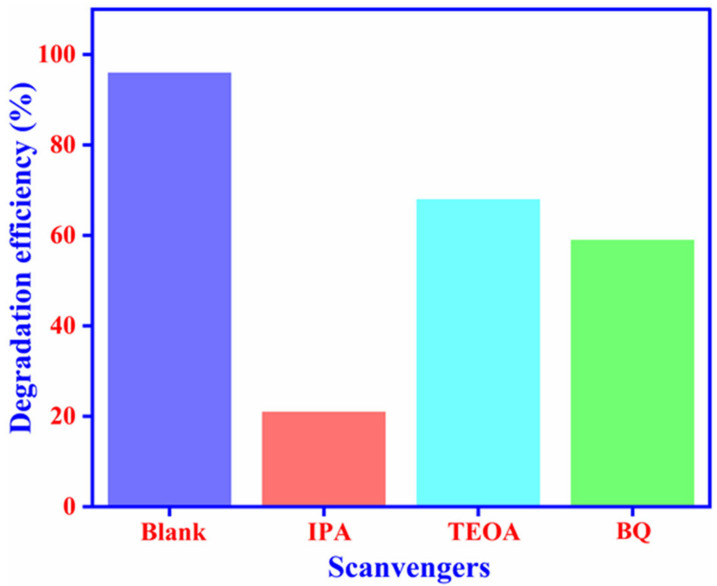
The effect of trapping experiment using different scavengers.

**Figure 10 ijms-23-04398-f010:**
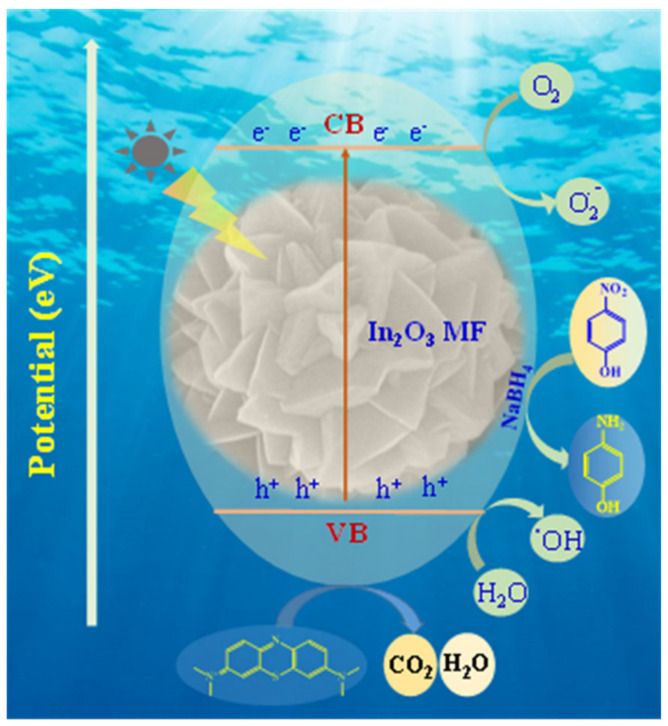
Proposed photocatalytic mechanism of reduction and degradation of 4-NP and MB by In_2_O_3_-MF catalyst.

## Data Availability

Not applicable.

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
