# Peer review of "A Flower-like In2O3 Catalyst Derived via Metal–Organic Frameworks for Photocatalytic Applications"

_ijms, 2022, doi:10.3390/ijms23084398_

Round 1

Reviewer 1 Report

The work submitted for publication in IJMS entitled “ A flower-like In2O3 catalyst derived via metal-organic frame-2 works for photocatalytic applications” by Munisamy Maniyazagan, Hyeon-Woo Yang, Perumal Naveenkumar, Nayoung Kang, Woo Seung Kang, Sun-Jae Kim describes the catalytic activity of a In based catalyst, using MOFs. It is not related to thermal catalysis but photocatalytic ones.

In the text, the authors should clarify more how the nanoparticles really are the active agents of the catalysis, thus, to remark which is the role of each agent. In those studies, always it is a concern to know if really the catalyst is the active agent, and even more difficult to know the structural motif that supports its activity.

In the conclusions remove “Finally, “; correct “The photocatalysts were evaluating”

Apart from some odd expressions and/or typos, and mainly clarify the blank tests, the paper could be considered for publication in IJMS.

Author Response

Response to Reviewer 1 Comments

Point 1: In the text, the authors should clarify more how the nanoparticles really are the active agents of the catalysis, thus, to remark which is the role of each agent. In those studies, always it is a concern to know if really the catalyst is the active agent, and even more difficult to know the structural motif that supports its activity.

Response 1: Thanks for the reviewer’s valuable comments. The preparation of mixed-ligand MOFs with multiple functions lining the cavities is emerging as a highly promising approach toward the implementation of fine-tuned reactivity. However, when mixed ligands were used, the difference in coordination chemistry between the ligands may result in different networks. Interpenetration of these different networks enriches the types of structurally interpenetrated systems.

Point 2: In the conclusions remove “Finally, “; correct “The photocatalysts were evaluating”

Response 2: Thank you for these important inputs. We have now modify the conclusion sentence.

Point 3: Apart from some odd expressions and/or typos, and mainly clarify the blank tests, the paper could be considered for publication in IJMS.

Response 3: Thank you for this comment. We have now improved the manuscript.

Reviewer 2 Report

Review of the manuscript ijms-1659617-v1- to Authors: This paper presents a novel synthesis of a flower like indium oxide metal organic framework catalyst.

Abstract - Please define all abbreviations when introducing them for the first time, i.e. MR and MD. Also, rewrite the abstract so it’s easier to follow, it’s a bit jumbled at the moment.

Introduction – Again like in the abstract you go from one thing to another, so please rewrite it more comprehensibly. “Furthermore, due to its low cost and great effectiveness, heterogeneous is an appealing technology for degrading organic pigments” Also please upgrade the language a bit.

“The status of their properties typically determines the variety in their attributes (crystalline or amorphous).” You are talking about In2O3 or a group of oxides??

Results and discussion – “Hard Lewis acids and bases, as well as mild Lewis acids and bases, are predicted to form strong coordination bonds. Hard bases, such as carboxylate-based ligands, can create stable MOFs with indium metal cations” maybe add references for this statements?

Fig1 could use an upgrade so the readers can clearly see the added species.

For the XRD again add a reference for the claim which peaks are attributed to the MOF structure please.

“Due to the presence of strong peaks in the XRD pattern, these In2O3-MD exhibit good crystallinity.” Please elaborate this sentence.

“Additionally, the crystallinity of the In2O3-MF is indicated by the distinct and strong peaks” Which peaks?

“Reflections from the cubic and rhombohedral In2O3 Bragg planes” Please explain this, and elaborate on the similarity of your peaks and ICDD card for In2O3?

“rankly micropetals” Meaning of this description??

“he micropetals are transferred to be disordered and porous structures.” Elaborate this sentence.

As for the UV/VIS and BET analysis in my opinion it would be wiser to include them in the manuscript and not the supplementary since these attributes are of greater importance for the photocatalytic properties than the XPS survey.

Fig 6 and 7 are a bit low in resolution. Also the whole discussion for the photocatalytic testing should be rewritten more clearly.

Also please add Fig 7 in the corresponding subsection, that is 2.2.2.

Materials and methods – Please write the techniques in the order they appear in the results and discussion section.

Conclusions – Ok, maybe just remove the finally from the first sentence since it’s redundant.

Almost all parts of the article need to be rewritten more concisely and clearly. Language and style also need a lot of upgrade. Some figures (order and resolution) also really need to be upgraded before publication. Finally, even though the synthesis and the demonstrated photocatalytic effectiveness are quite compelling and interesting for this field I must recommend major revision.

Author Response

Response to Reviewer 2 Comments

Point 1: Abstract - Please define all abbreviations when introducing them for the first time, i.e. MR and MD. Also, rewrite the abstract so it’s easier to follow, it’s a bit jumbled at the moment.

Response 1: Thank you for this comment. Now we can cange the abstract section.

Point 2: Introduction – Again like in the abstract you go from one thing to another, so please rewrite it more comprehensibly. “Furthermore, due to its low cost and great effectiveness, heterogeneous is an appealing technology for degrading organic pigments” Also please upgrade the language a bit.

Response 2: Thank you for this comment. Introdcution part we have carfully checked and its okay, and rewrite the above sentence.

Point 3: “The status of their properties typically determines the variety in their attributes (crystalline or amorphous).” You are talking about In2O3 or a group of oxides??

Response 3: Thank you for this comment. General group of oxides.

Point 4: Results and discussion – “Hard Lewis acids and bases, as well as mild Lewis acids and bases, are predicted to form strong coordination bonds. Hard bases, such as carboxylate-based ligands, can create stable MOFs with indium metal cations” maybe add references for this statements?

Response 4: Thank you for this comment. Now we can add the reference for this statement.

Point 5: Fig1 could use an upgrade so the readers can clearly see the added species.

Response 5: Thank you for this comment. Now we can redraw the Fig.1.

Point 6: For the XRD again add a reference for the claim which peaks are attributed to the MOF structure please.

Response 6: Thank you for this comment. Now we iclude the reference for MOF structure.

Point 7: “Due to the presence of strong peaks in the XRD pattern, these In2O3-MD exhibit good crystallinity.” Please elaborate this sentence.

Response 7: Thank you for this comment. Now we can change the sentence.

Point 8: “Additionally, the crystallinity of the In2O3-MF is indicated by the distinct and strong peaks” Which peaks?

Response 8: Thank you for this comment. 2q = 17.6°, 18.7°, 27.1°, and 33.3°, respectively

Point 9: “Reflections from the cubic and rhombohedral In2O3 Bragg planes” Please explain this, and elaborate on the similarity of your peaks and ICDD card for In2O3?

Response 9: Thank you for this comment. Bragg planes (104), (110), (024), (116), and (214) were recorded at 2q = 30.0°, 33.3°, 46.1°, 51.6°, and 57.4°, respectively. The diffraction pattern is extremely similar to that of In2O3 (JCPDS file no. 21-0406

Point 10: “rankly micropetals” Meaning of this description??

Response 10: Thank you for this comment. Micropetals means the flower-like structure having more petals, and the size was micro structure. For that reason we said micropetals. Before annealing the micropetals having no porous structure once annealed its disorderd and porous structure. It clearly explin in SEM and TEM images.

Point 11: “he micropetals are transferred to be disordered and porous structures.” Elaborate this sentence.

Response 11: Thank you for this comment. Micropetals means the flower-like structure having more petals, and the size was micro structure. For that reason we said micropetals. Before annealing the micropetals having no porous structure once annealed its disorderd and porous structure. It clearly explin in SEM and TEM images.

Point 12: As for the UV/VIS and BET analysis in my opinion it would be wiser to include them in the manuscript and not the supplementary since these attributes are of greater importance for the photocatalytic properties than the XPS survey.

Response 12: Thank you for this comment. UV band gap values and BET surface area values are given in mauscript and more figures in the present mauscript. So we given in supporting information.

Point 13: Fig 6 and 7 are a bit low in resolution. Also the whole discussion for the photocatalytic testing should be rewritten more clearly.

Response 13: Thank you for this comment. Now we have revised the Fig. 6 and Fig. 7. We are 600 dpi resolution figure only used in this maunscript.

Point 14: Also please add Fig 7 in the corresponding subsection, that is 2.2.2.

Response 14: Thank you for this comment. We have now chenge the Fig. 7 in subsection 2.2.2.

Point 15: Materials and methods – Please write the techniques in the order they appear in the results and discussion section.

Response 15: Thank you for this comment. We have now improved the Materials and methods section.

Point 16: Conclusions – Ok, maybe just remove the finally from the first sentence since it’s redundant.

Response 16: Thank you for this comment. We have now correct the sentence.

Reviewer 3 Report

This article with No. ijms-1659617 shows complete study in catalyst (In2O3 system) characterization and photocatalytic performance. In fact, this article has achieved the level of acceptance. When fulfilling below points, this article can become acceptable.

Some points still be re-filled by authors and they are listed as below:

  1. In figure 2, the description of (b) should be added
  2. This study shows the flower-like In2O3 catalyst derived via metal-organic frameworks but the authors seem to forget to write the reagents to form metal-organic frameworks. The authors should add these reagents.
  3. Finally, recommended the authors to draw diagram containing the MOF (metal-organic frameworks) reagents bonded to the In2O3 (maybe In2O3 surface).

Author Response

Response to Reviewer 3 Comments

Point 1: In figure 2, the description of (b) should be added

Response 1: Thank you for this comment. Now we have include the description in Fig.2.

Point 2: This study shows the flower-like In2O3 catalyst derived via metal-organic frameworks but the authors seem to forget to write the reagents to form metal-organic frameworks. The authors should add these reagents.

Response 2: Thank you for this comment. We have include the orhanic ligands in synthesis scheme and mateials and methods.

Point 3: Finally, recommended the authors to draw diagram containing the MOF (metal-organic frameworks) reagents bonded to the In2O3 (maybe In2O3 surface).

Response 3: Thank you for this comment. We have redraw the diagram. But exact bonding and MOF structurer very dificult to draw. Because we did not get any single crystal or computaional analysis. We consider the comments and will give details studies in future manuscripts.
